# ADAPTIVE SPARSE ALLOCATION WITH MUTUAL CHOICE & FEATURE CHOICE SPARSE AUTOENCODERS

## ABSTRACT

Sparse autoencoders (SAEs) are a promising approach to extracting features from neural networks, enabling model interpretability as well as causal interventions on model internals. SAEs generate sparse feature representations using a sparsifying activation function that implicitly defines a set of token-feature matches. We frame the token-feature matching as a resource allocation problem constrained by a total sparsity upper bound. For example, TopK SAEs solve this allocation problem with the additional constraint that each token matches with at most $k$ features. In TopK SAEs, the $k$ active features per token constraint is the same across tokens, despite some tokens being more difficult to reconstruct than others. To address this limitation, we propose two novel SAE variants, *Feature Choice SAEs* and *Mutual Choice SAEs*, which each allow for a variable number of active features per token. Feature Choice SAEs solve the sparsity allocation problem under the additional constraint that each feature matches with at most $m$ tokens. Mutual Choice SAEs solve the unrestricted allocation problem where the total sparsity budget can be allocated freely between tokens and features. Additionally, we introduce a new auxiliary loss function, `aux_zipf_loss`, which generalises the `aux_k_loss` to mitigate dead and underutilised features. Our methods result in SAEs with fewer dead features and improved reconstruction loss at equivalent sparsity levels as a result of the inherent adaptive computation. More accurate and scalable feature extraction methods provide a path towards better understanding and more precise control of foundation models.

## 1 INTRODUCTION

Understanding the internal mechanisms of neural networks is a core challenge in Mechanistic Interpretability. Increased mechanistic understanding of foundation models could provide model developers with tools to identify and debug undesirable model behaviour.

Dictionary learning with sparse autoencoders (SAEs) has recently emerged as a promising approach for extracting sparse, meaningful, and interpretable features from neural networks, particularly language models (Huben et al., 2024; Sharkey et al., 2022).

One problem with wide SAEs for foundation models is that there are often many dead features (Rajamanoharan et al., 2024a; Templeton et al., 2024; Gao et al., 2024). Dead features are features which are remain inactive across inputs, effectively wasting model capacity and hampering efficient training. Another problem is that approaches like TopK SAEs (Gao et al., 2024) don't have a natural way to take advantage of Adaptive Computation: spending more computation, and crucially more features, to reconstruct more difficult tokens.

We frame the problem of generating sparse feature activations corresponding to some given neural activations as a resource allocation problem, allocating the scarce total sparsity budget between token-feature matches to maximise the reconstruction accuracy. Within this framing, we naturally motivate two novel SAE variants which can provide Adaptive Computation: Feature Choice SAEs and Mutual Choice SAEs (FC and MC SAEs respectively). Feature Choice SAEs solve the sparsity allocation problem under the additional constraint that each feature matches with at most m tokens. Mutual Choice SAEs solve the unrestricted allocation problem where the total sparsity budget can be allocated freely between tokens and features. These approaches combine the Adaptive Computation of Standard SAEs with the simple optimisation and improved performance of TopK SAEs.

Our contributions are as follows:

- We provide a framing for sparsifying activation functions in SAEs as a solution to a resource allocation problem.

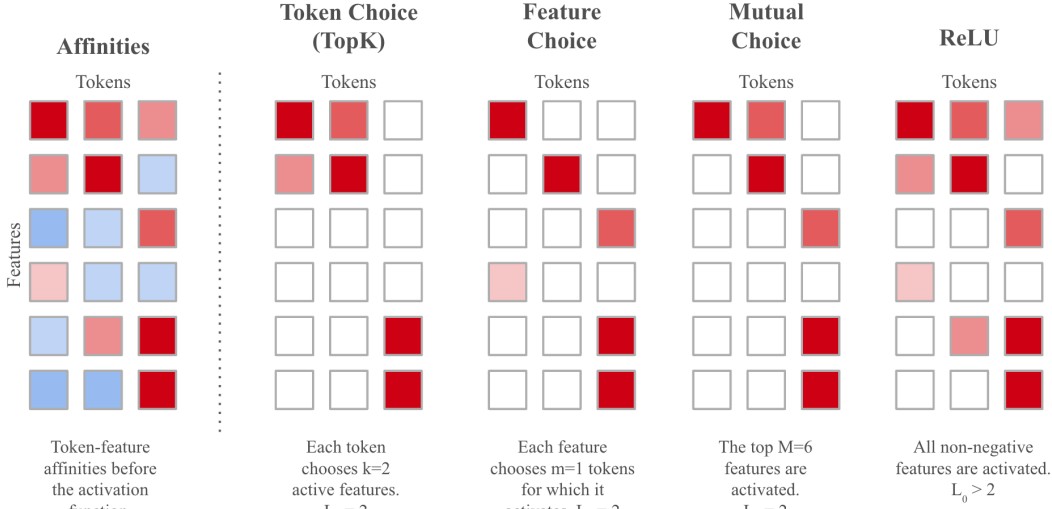

Figure 1: A comparison of the pre-activation affinities and the resulting feature activations following different sparsifying activation functions. Red and blue represent positive and negative affinities respectively, with deeper colours representing larger magnitudes. In the first three approaches we have a total sparsity budget of 6. *Affinities (Far-left)*: The token-feature affinities $\mathbf{Z}$', before any sparsifying activation function. *Token Choice/TopK (Center-left)*: We activate the top $k$ features corresponding to each token. Note that there are features that don't fire in this batch, which could lead to dead features. *Feature Choice (Center)*: For each feature, it activates corresponding to the top $m$ tokens with the highest affinity. Note that all features fire in this batch. *Mutual Choice (Center-right)*: The elements with the largest magnitude affinities activate, regardless of their token or feature affiliations. *ReLU/Standard (Far-right)*: All strictly positive elements activate. Here we allow low-magnitude feature activations which may be false positives and which cause the $L_0$ to be higher.

- We introduce two new SAE architectures: the **Mutual Choice SAE** and **Feature Choice SAE**, which are Pareto improvements on both standard SAEs and TopK SAEs. Additionally, the Feature Choice approach is, to our knowledge, the first SAE training method which reliably results in *zero dead features* even at large scale.
- We show that our methods naturally enable *Adaptive Computation*: using more features to reconstruct more difficult tokens. Instead of setting the number of features per token as a fixed $k$, we fix $\mathbb{E}[k]$ as a hyperparameter and allow the model to learn how to allocate the sparsity budget, without increasing computational overhead.
- We propose a novel auxiliary loss function, `aux_zipf_loss`, which mitigates under-utilisation of features and better utilises the SAE's full capacity.

We open-source a reference implementation for the community at [REDACTED]. We believe that with increasingly accurate approaches to feature extraction, it will become possible to connect sparse features over many layers and understand how models are computing outputs in a mechanistic, circuits-driven fashion. In particular, given that Feature Choice SAEs generally have no dead features, they can scale reliably to very large autoencoders, which are likely to be necessary for effective reconstruction on large foundation models such as GPT-4 (OpenAI et al., 2024) or Llama 3 (Dubey et al., 2024).

## 2 RELATED WORK

### 2.1 SPARSE AUTOENCODERS

**Sparse Autoencoders (SAEs)** (Lee et al., 2007; Le, 2013; Mairal et al., 2014) learn an over-complete basis, or dictionary, of sparsely activating features. The feature activations, $\mathbf{z}$, correspond to their associated neural activations, $\mathbf{x}$, via the feature dictionary. In particular, we can write an SAE as:

$$\mathbf{z} = \sigma_s(\text{Enc}(\mathbf{x})) \in \mathbb{R}^F \tag{1}$$

$$\hat{\mathbf{x}} = \text{Dec}(\mathbf{z}) \in \mathbb{R}^N \tag{2}$$

where $\sigma_s$ is a sparsifying activation function (e.g. ReLU), Dec is an affine map and $\mathbf{x} \in \mathbb{R}^N$ [1]. $\mathbf{z}' = \text{Enc}(\mathbf{x})$ are the pre-activation features, which we will call the token-feature *affinities*.

SAEs are trained to minimize the Reconstruction Error (Mean Squared Error) between $\mathbf{x}$ and $\hat{\mathbf{x}}$. This reconstruction error term is combined with an optional Sparsity Loss term (for example, an $L_1$ penalty to induce sparsity) and an optional Auxiliary Loss term to reduce dead features:

$$\mathcal{L}(\mathbf{x}) = |\mathbf{x} - \hat{\mathbf{x}}|_2^2 + \lambda_1 \mathcal{L}_{\text{sparsity}}(\mathbf{z}) + \lambda_2 \mathcal{L}_{\text{aux}}(\mathbf{x}, \mathbf{z}, \hat{\mathbf{x}}) \tag{3}$$

Rajamanoharan et al. (2024a); Templeton et al. (2024); Gao et al. (2024) have shown that decomposing neural activations using the SAE feature dictionary allows for increased human interpretability of models even at model sizes comparable to frontier foundation models.

## 2.2 ADAPTIVE COMPUTATION

In **Adaptive Computation**, neural networks decide how much compute (and/or which parameters) to apply to a given input example (Graves, 2017; Xue et al., 2023). Ideally, the model should learn to apply less compute to easier examples and more compute to more difficult examples in order to maximise performance within a compute budget. In our setting, we consider the token-feature matches to be the scarce quantity to allocate, where we say that a token matches with a feature if the feature is activated on that token.

## 2.3 TOPK SAEs

**TopK SAEs** (Gao et al., 2024) use a TopK activation function instead of the $L_1$ penalty to induce sparsity, as in Makhzani & Frey (2014). Though in the standard $L_1$ SAE formulation, the number of features-per-token is variable, in the TopK formulation the features-per-token is fixed at the same $k$ for all tokens. We hypothesize that having a fixed $k$ is a key drawback of the TopK method. Variable $k$ values introduce Adaptive Computation which can focus more of the token-feature matching budget on more difficult tokens.

In concurrent work, Bussmann et al. (2024) introduce BatchTopK which is closely analogous to our Mutual Choice SAEs and also provides adaptive computation. However, they do not deal with the problem of underutilised features.

## 2.4 DEAD FEATURES

SAE features which remain inactive across many inputs are known as **dead features**. Bricken et al. (2023) declare a feature to be dead when it hasn't fired for at least 1e7 tokens. Dead features present a challenge especially when scaling to larger models and wider autoencoders. For example, Templeton et al. (2024) find 64.7% of features are dead for their autoencoders with 34M features. Our Feature Choice approach naturally results in *zero dead features* by ensuring that each feature activates for every batch.

## 2.5 AUXILIARY k LOSS FUNCTION

Gao et al. (2024) propose the auxiliary loss function, `aux_k_loss` to reduce the proportion of dead features. Given the SAE residual $\mathbf{e} = \mathbf{x} - \hat{\mathbf{x}}$, they define the auxiliary loss $\mathcal{L}_{\text{aux}} = |\mathbf{e} - \hat{\mathbf{e}}|^2$, where $\hat{e} = \text{Dec}(z_{\text{dead}})$ is the reconstruction using the top $k_{\text{aux}}$ dead features. Gao et al. (2024) find fewer dead features when using the `aux_k_loss` for SAE training.

However, the `aux_k_loss` is only applied to features which qualify as dead. We apply a similar auxiliary loss, the `aux_zipf_loss`, to underutilised, but not yet dead, features.

## 3 BACKGROUND

### 3.1 SPARSIFYING ACTIVATION FUNCTIONS AS RESOURCE ALLOCATORS

We consider the following many-to-many matching problem:

- We have $F$ features and a batch of $B$ tokens. We would like to have at most $M$ token-feature matches, where we say that a token matches with a feature if the feature is activated on that token [2].

---

[1] In practice, there may be additional pre-processing and post-processing of the neural activations, $\mathbf{x}$.

[2] Here we follow the Mixture of Expert literature in abusing notation slightly to refer to neural activations corresponding to a given token position as a "token".

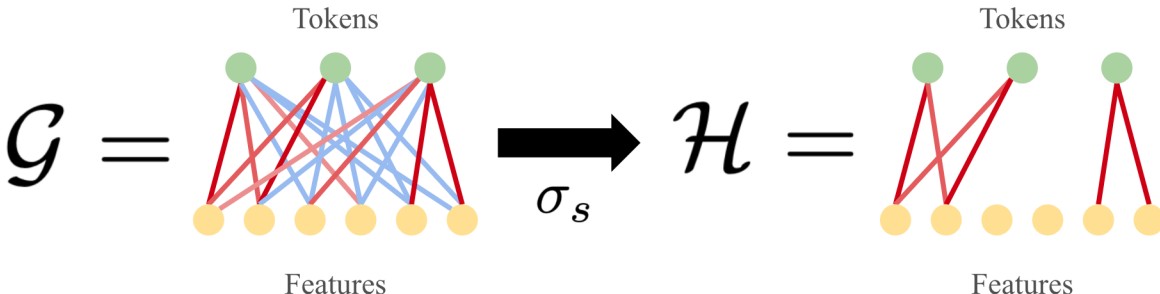

Figure 2: An illustration of the sparsifying activation function $\sigma_s$ acting on the token-feature affinities. $\mathcal{G}$ (left) is a weighted bipartite graph $\mathcal{G} = \{\{T_1, T_2, T_3\} \times \{F_1, F_2, ..., F_6\}, \mathbf{E}\}$. Edge weights represent token-feature affinities, with red and blue representing positive and negative values respectively. We are seeking a subgraph $\mathcal{H} \subseteq \mathcal{G}$ with $M = 6$ edges. Here we have defined $\mathcal{H}$ (right) by the TopK method for $k = 2$; we select the 2 edges from each token with the largest edge weights. This provides an equivalent view to Figure 1 in terms of bipartite graphs.

- We would like to allocate our budget of $M$ token-feature matches such that the reconstruction error is minimised.

Formally, we seek a reconstruction-error optimal weighted subgraph $\mathcal{H} \subseteq \mathcal{G} = \{\{1, ..., B\} \times \{1, ..., F\}, \mathbf{E}\}$ where $\mathcal{H}$ has at most $M$ edges, for $M \ll BF$. Figure 2 details an example of a given graph $\mathcal{G}$ and subgraph $\mathcal{H}$. The edge weights can be viewed as the token-feature affinities: the pre-sparsifying activation feature magnitudes $\mathbf{z}'$. [3]

This problem doesn't immediately admit an efficient solution because it is currently unspecified how the edge weights contribute to the token reconstruction error. We make a simplifying assumption that we denote the **Monotonic Importance Heuristic** - the edges with the largest edge weights are likely to represent the most important contributions to the reconstruction error. With this heuristic, we can solve the problem of allocating token-feature matches by choosing the $M$ edges with the largest magnitude edge weights as the edges for our subgraph $\mathcal{H}$.

We can equivalently view this allocation problem as choosing a binary mask $\mathbf{S} \in \mathbb{R}^{B \times F}$ with at most $M$ non-zero elements which maximises reconstruction accuracy. This mask is to be element-wise multiplied with a token-feature affinity matrix $\mathbf{z}' \in \mathbb{R}^{B \times F}$. Applying the Monotonic Importance Heuristic, we are looking for the mask $\mathbf{S}$ such that $\sum_{i,j} \mathbf{z}'_{i,j} = \sum_{i,j} \mathbf{z}'_{i,j} \odot \mathbf{S}_{i,j}$ is maximised.

**TopK SAEs:** We can now see the TopK SAE approach as a special case of the above allocation problem, with the additional constraint that the number of features per token is at most $k$ for each token. In other words, $\sum_i (\mathbf{S}_{i,j}) = k \, \forall j$, where $M = kB$. This leads to the solution of $\mathbf{S} = \text{TopKIndices}(\mathbf{z}', \dim = -1)$, i.e. $\mathbf{S}$ picks out the $k$ features with the highest affinity for each token [4]. Here $\sigma_s(\mathbf{z}') = \mathbf{S} \odot \mathbf{z}'$; element-wise multiplication with $\mathbf{S}$ defines our sparsifying activation function $\sigma_s$.

We now consider two other variants of this problem displayed in Figure 1:

**Feature Choice SAEs:** Whilst TopK SAEs require each token to match with at most $k$ features, we instead add the constraint that each feature matches with at most $m$ tokens. This is equivalently a constraint on the columns of S rather than its rows: $\sum_j (\mathbf{S}_{i,j}) = m \, \forall i$, where $M = mF$. This leads to the solution of $\mathbf{S} = \text{TopKIndices}(\mathbf{z}', \dim = 0)$, i.e. $\mathbf{S}$ picks out the $m$ tokens with the highest affinity for each feature.

**Mutual Choice SAEs:** Here we don't add any additional constraints and allow any choice of token-feature matching. This leads to the solution of $\mathbf{S} = \text{TopKIndices}(\mathbf{z}', \dim = (0, 1))$, i.e. $\mathbf{S}$ picks out the largest elements of the $\mathbf{z}'$ affinity matrix, regardless of their position.

The Feature Choice (FC) and Mutual Choice (MC) sparsifying activation functions can be seen as having the desirable properties of the TopK activation (for example, preventing activation shrinkage, reducing the impact of noisy, low magnitude activations, allowing for a progressive recovery code, enabling simple model comparison, not requiring sparsity losses which are in conflict with the reconstruction loss etc.) but whilst allowing for Adaptive Computation, see

---

[3]As an illustrative example of the same problem, we can imagine a university which is able to confer at most $M$ degrees where a student may take many degree subjects and a degree subject may admit many students.

[4]See Appendix H for a proof that this is the optimal solution under the constraints.


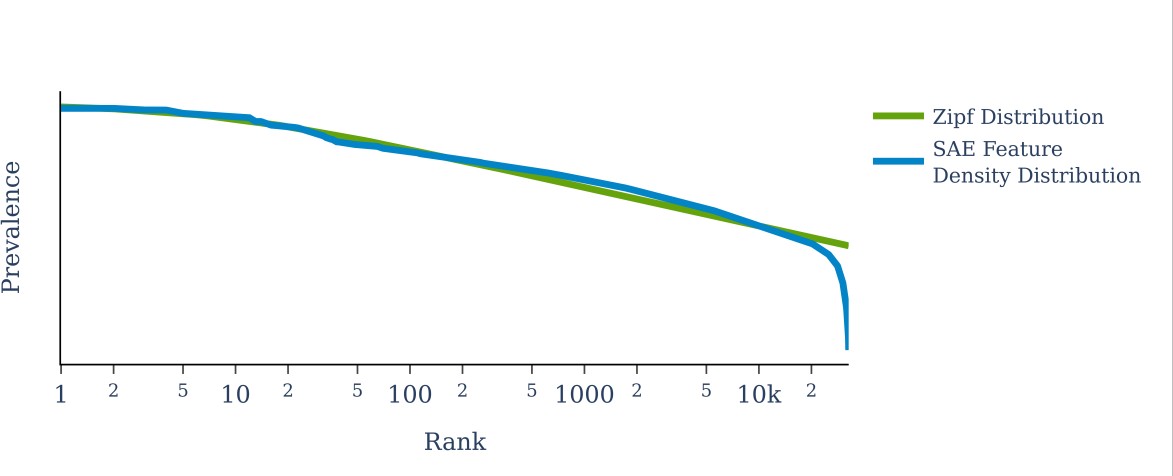

Figure 3: The Feature Density Curve fits a Zipf curve with $R^2 = 0.982$. The middle segment of the feature density distribution (features 100-20,000) fit the Zipf curve with $R^2$=0.996

Figure 1. Since we do not have constraints on the number of features per token in FC or MC, it's possible for one token to activate (i.e. match with) more features than another token.

### 3.1.1 ANALOGY TO MIXTURE OF EXPERTS ROUTING

We choose our naming here to align with the study of token-expert matching in the Mixture of Experts paradigm, where there is a close analogy. Token Choice MoE routing strategies (Shazeer et al., 2017; Fedus et al., 2022) have the constraint that each token can be routed to at most $k$ experts allowing for expert imbalances. On the other hand, Expert Choice routing strategies (Zhou et al., 2022) have the constraint that each expert processes at most m tokens, which eliminates the possibility of underutilised experts and allows tokens to be routed to a variable number of experts.

Here the intuitions are "*each token picks the $k$ experts to be routed to*" and "*each expert picks the $m$ tokens to be routed to that expert*" for Token Choice and Expert Choice respectively. The variety of MoE routing algorithms is explored in Liu et al. (2024). The *Feature Choice* approach we propose is directly analogous to the *Expert Choice* approach in MoEs and *TopK* SAEs are directly analogous to *Token Choice* MoEs. For this reason, we will call TopK SAEs, Token Choice SAEs to unify our notation.

### 3.2 DISTRIBUTION OF FEATURE DENSITIES

When analysing the distribution of feature densities in open-source SAEs from Gao et al. (2024), we find (as shown in Figure 3) that the distributions typically follow a power law described by the Zipf distribution with $R^2 = 0.982$.

We note that although the Zipf distribution well fits most of the distribution, there is a considerable residual at the lower end of the distribution (the 20k+ rank tokens). We suggest that these features are underutilised. Underutilised features see fewer gradient updates than other features leading to a self-reinforcing cycle of these features being less useful and hence further underutilised until they die.

We will refer to these underutilised features as **dying features**. We define a dying feature as a feature which is one of the 25% least prevalent features which is also <60% as prevalent as we might predict from the prevalence rank of the feature using the fitted Zipf curve. More formally:

**Definition 3.1.** *Given a feature set* $\mathcal{F} = \{\mathbf{f_1}, \mathbf{f_2}, ..., \mathbf{f_F}\}$ *ordered by feature prevalence[5], we say* $\mathbf{f_i}$ *is a **dying feature** if:*

1. $i > \frac{3}{4}F$ - *i.e. is in the bottom quartile of features when ranked by prevalence*

2. $\frac{actual\ prevalence(\mathbf{f_i})}{expected\ prevalence(\mathbf{f_i})} \equiv \frac{actual\ prevalence(\mathbf{f_i})}{Zipf(i)} < 0.6$

---

[5]That is feature $\mathbf{f_i}$ is the i'th most prevalent feature

*3. $\forall j > i$, $\mathbf{f_j}$ is also a dying feature*

Previous approaches to dealing with dead features either resampled dead features (Bricken et al., 2023) or applied gradients to dead features (for example `aux_k_loss` (Gao et al., 2024)) but they didn't address dying features. We hypothesise that many of the revived dead features were still not appropriately utilised.

## 4 METHODS

### 4.1 MUTUAL CHOICE AND FEATURE CHOICE ACTIVATION FUNCTION

As detailed in Section 3.1, we introduce two activation functions for SAEs: the *Mutual Choice and Feature Choice TopK activation functions*. Our SAEs have the same structure as standard (ReLU) SAEs and TopK SAEs except for in activation function:

- **Mutual Choice Activation Function**: $\mathbf{z} = \sigma_s(\mathbf{z}') = \text{TopK}(\mathbf{z}', k = M, \dim = (0, 1))$;
- **Feature Choice Activation Function**: $\mathbf{z} = \sigma_s(\mathbf{z}') = \text{TopK}(\mathbf{z}', k = m_i, \dim = 0)$;

### 4.2 REDUCING THE PROPORTION OF DYING FEATURES

To address the problem of dying features discussed in Section 3.2, we add an additional auxiliary loss for dying features, which is a natural generalisation of the `aux_k_loss`. Given the SAE residual $\mathbf{e} = \mathbf{x} - \hat{\mathbf{x}}$, we define the auxiliary loss $\mathcal{L}_{\text{aux\_zipf}} = |\mathbf{e} - \hat{\mathbf{e}}|^2$, where $\hat{\mathbf{e}} = \text{Dec}(\mathbf{z}_{\text{dying}})$ is the reconstruction using the top $k_{\text{aux}}$ dying features. We can think of this `aux_zipf_loss` acting *preventatively* on features which could be at risk of becoming dead and acting *rehabilitatively* on features which have been recently revived. In this way, we reduce the proportion of both dead and dying features.

### 4.3 CHOOSING THE FEATURE CHOICE CONSTRAINT

In the Feature Choice approach, there remains the question of how to distribute the sparse feature activations across the feature dimension.

The simplest approach to this is to take $m_i = M/F$ for all *i*, where *F* is the number of features. In other words, each feature can pick exactly *m* tokens to process. We call this approach *Uniform Feature Choice*.

Uniform prevalence is a natural way to organize the features so that a feature firing provides maximal information about the token, under the assumption that all features provide approximately equal information. However, we have seen that in existing open-source SAEs, all features are not equivalently prevalent. Instead, they are approximately Zipf-law distributed. To maintain this distribution of feature density we choose $m_i \sim \text{Zipf}(\alpha, \beta)$, where Zipf represents a truncated Zipf distribution and *i* is the rank of a given feature in terms of feature density.

$$m_i = \text{Zipf}(i) \propto \frac{1}{(i + \beta)^\alpha} \tag{4}$$

We call the Feature Choice approach where the $m_i$ are Zipf-distributed, *Zipf Feature Choice*, henceforth simply *Feature Choice*.

### 4.4 TRAINING APPROACH

Our approach is as follows:

- Given the Zipf exponent and bias hyperparameters, $\alpha$ and $\beta$, [6], we use Algorithm 1 to determine the estimated feature density for each ranked feature. We use the estimated feature densities to define the threshold for dying features for the `aux_zipf_loss`.

---

[6]We may obtain these hyperparameters by performing a hyperparameter sweep. Alternatively, if we have trained SAE of the same dimensions, we may run inference with this SAE over a large dataset (100 million tokens) and track the number of times each feature activates as its *feature density* and each feature's relative feature density *rank*. We can then fit the (rank, feature density) pairs to a Zipf distribution and estimate the exponent and bias parameters, $\alpha$ and $\beta$. For GPT-2 sized residual stream activations (n=768), we find $\alpha \approx 1.0$, $\beta \approx 6.8$. In our experiments, we fix the exponent to exactly 1 for simplicity and we find that we may reuse similar $\alpha$ and $\beta$ parameters across SAE widths.

- We then train Mutual Choice SAEs with both the `aux_zipf_loss` and `aux_k_loss`.

- Finally, we optionally fine-tune these SAEs with the Feature Choice activation function, adding the constraint on the number of tokens that each feature should process. Here there are no auxiliary loss terms.

The sparsifying activation functions $\sigma_s$ for each approach are all TopK activations where the TopK is taken over the feature dimension for Token Choice (i.e. TopK), the batch dimension for Feature Choice and all dimensions for Mutual Choice.

## 5 EXPERIMENTAL SETUP

**Inputs:** We train our sparse autoencoders on the layer 6 residual stream activations of GPT-2 small (Radford et al., 2019). For larger SAE widths (over 1M features) we train our sparse autoencoders on the 24th layer residual stream activations of Pythia-2.8B-deduped (Biderman et al., 2023). We use a context length of 64 tokens for all experiments. We preprocess the activations by subtracting the mean over the $d_{\text{model}}$ dimension and normalize all inputs to unit $L_2$ norm. All experiments use the FineWeb dataset (Penedo et al., 2024) unless otherwise specified. We shuffle the activations for training our SAEs (as in Nanda (2023)). Experiments with Token Choice, Mutual Choice and Feature Choice SAEs are performed without feature resampling, where Standard and SAE++ SAEs are trained with resampling [7].

**Hyperparameters:** We tune learning rates based on Gao et al. (2024) suggestion that the learning rate scales like $\sqrt{n}$. We use the AdamW optimizer (Kingma & Ba, 2015) and a batch size of 1,536 [8]. We train each SAE for 10,000 steps or until convergence. We use a weight decay of 1e-5 and apply gradient clipping. We analyse SAE with widths from 4x to 32x larger than the size of the $d_{\text{model}}$ dimension. We use gradient accumulation for larger batch sizes. We do not perform extensive hyperparameter sweeps.

**Evaluation:** After training, we evaluate autoencoders on sparsity $L_0$, reconstruction (MSE) and the difference on the model's final (Cross-Entropy) loss. We report a standard normalized version of the loss recovered (%). We additionally evaluate our SAEs' interpretability using Juang et al. (2024)'s automated interpretability (AutoInterp) process. We report the percentage of dead features across models.

**Baselines:** We compare our SAEs against Standard (ReLU) SAEs and TopK SAEs [9].

## 6 RESULTS

We find that Feature Choice SAEs are a Pareto improvement upon the Token Choice TopK SAEs, as in Figure 4. Similarly Figure 5 illustrates that both Mutual Choice and Feature Choice SAEs provide better utilisation of the SAE capacity with fewer dead features than comparable SAE methods. Notably the Feature Choice SAE method results in the fewest (often zero) dead features.

| SAE Type | 6k latent dim | 16m dim | 34m dim |
|---|---|---|---|
| Standard SAE (w/ resampling) | 9.0% | >90.0% | – |
| SAE++ (w/ resampling) | 5.1% | – | 64.7% |
| Token Choice (TopK) SAE | 0.0% | 7.0% | – |
| Feature Choice SAE (Ours) | **0%** | **0%** | **0%** |

Table 1: The Feature Choice SAE maintains zero dead features at widths of up to 34 million features. This is in contrast to Standard SAEs (Bricken et al., 2023), the SAE++ (Templeton et al., 2024) and Token Choice (TopK) SAE (Gao et al., 2024) which have an increasing percentage of dead features with the SAE width.

---

[7]Bricken et al. (2023) describe a resampling procedure for dead features in which they reinitialise these dead features midway through training in order to reduce the number of dead features at the end of training. Models with the `aux_k_loss` do not require resampling but for models without the `aux_zipf_loss` we include resampling for stronger baselines.

[8]For the Feature Choice approach, it's important to have sufficiently large minibatch sizes so that each feature is expected to activate every few minibatches.

[9]We do not explicitly test against Gated SAEs, but Gao et al. (2024) find that TopK SAEs perform similarly or better than Gated SAEs with $1.5\times$ less compute to convergence.

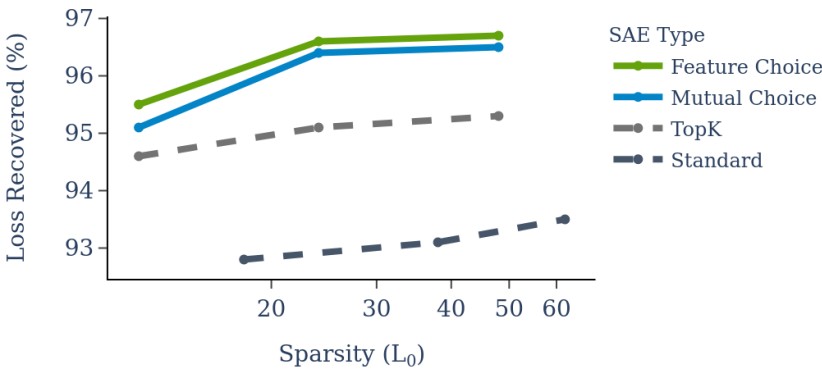

Figure 4: SAEs trained with the Mutual Choice activation function, and those finetuned with the Feature Choice activation function have up to 1.7% greater normalised reconstruction loss recovered at equivalent sparsity levels compared to TopK and standard SAEs.

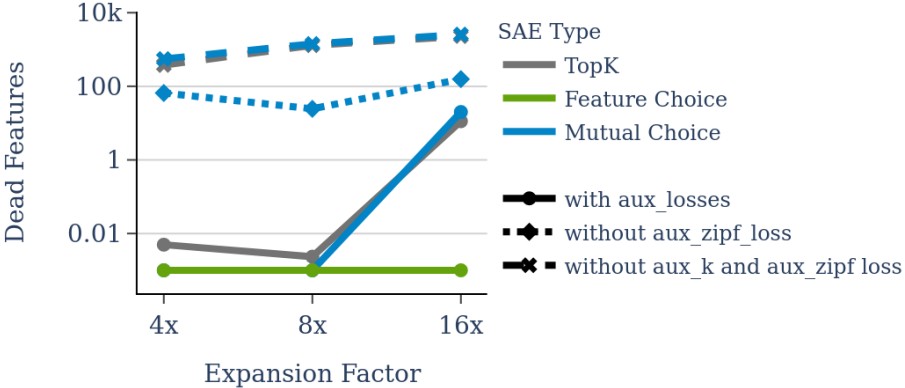

Figure 5: Both Mutual Choice SAEs and Feature Choice SAEs have fewer dead features than SAEs trained without the `aux_zipf_loss`. Even at $16\times$ expansion factor, Feature Choice SAEs have no dead features. The `aux_zipf_loss` and `aux_k_loss` are effective at reducing the number of dead features in Mutual Choice SAEs. At large SAE widths, the `aux_zipf_loss` reduces the number of dead features by up to $13\times$.

Templeton et al. (2024)'s 34 million latent SAEs have a dead feature rate of 64.7% (with resampling); Gao et al. (2024)'s 16 million latent SAEs have a dead feature rate of 90% without mitigations and 7% with mitigations. We find that Feature Choice SAEs can be trained with a 0% dead feature rate for both 16m and 34m latent SAEs as in Table 1.

## 7 DISCUSSION

We summarize the comparison between our approach and related approaches in Table 2:

To our knowledge, we present the first SAE training process which explicitly contains two separate phases: first *Mutual Choice training*, then *Feature Choice training*. We speculate that there may be performance benefits to phased training.

| SAE Type | Performance | Dead Features | Auxiliary Losses |
|---|---|---|---|
| Standard SAE | Weak | Many | $L_1$ Sparsity loss |
| SAE++ | Better | Many | Scaled $L_1$ sparsity loss |
| Jump-ReLU SAE | Better | Fewer | $L_1$ Sparsity loss |
| Token Choice (TopK) SAE | Better | Fewer | `aux_k` |
| BatchTopK SAE | Better | Fewer | `aux_k` |
| Mutual Choice SAE (Ours) | **Best** | **Fewer** | `aux_k` and `aux_zipf` |
| Feature Choice SAE (Ours) | **Best** | **None** | **None** |

Table 2: Mutual Choice and Feature Choice SAEs (ours) achieve higher reconstruction accuracy with fewer dead features than other approaches such as Standard SAEs (Bricken et al., 2023), the SAE++ (Templeton et al., 2024), Jump-ReLU SAEs (Rajamanoharan et al., 2024b), Token Choice (also known as TopK) SAEs (Gao et al., 2024) and BatchTopK (Bussmann et al., 2024).

We believe that conceiving of the role of SAE encoders as defining matching/routing algorithms (similar to other Adaptive Computation work, for example, within the Mixture of Experts literature) could be a valuable intuition pump for further improvements to SAE architectures.

Our approaches can also be combined with the MDL-SAE (Ayonrinde et al., 2024) formulation which treats **conciseness** (**Description Length**) as the relevant quantity for evaluation and model selection rather than sparsity ($L_0$). The Description Length of a set of feature activations is a function of the sparsity, the distribution of activation patterns for each feature and the SAE width.

The computational efficiencies of the Token Choice (TopK), Mutual Choice and Feature Choice approaches are theoretically equivalent in terms of Floating Point Operations (FLOps) and in our experiments we find empirically that the Mutual Choice and Feature Choice activation functions do not add significant computational requirements (<1% difference in wall clock time) despite converging more quickly.

### 7.1 ZIPF DISTRIBUTED FEATURES

We find that constraining the number of tokens per feature with the Zipf distribution outperforms using the uniform distribution by >10% model loss recovered. The large drop in performance using the uniform distribution compared to the Zipf distribution gives additional evidence that naturally occurring features are not uniformly distributed.

We hypothesise that the reason that features appear to be Zipf distributed may be strongly analogous to the reason that the frequency of words in natural languages like English are also Zipf distributed. Words are the semantic units of sentences. Since features are the semantic units of computation within language models, a similar mechanism could explain the empirical tendency for the densities of SAE features to tend towards the Zipf distribution.

In the Computational Linguistics literature, it is well known that the distribution of words in natural languages approximately follows a Zipf distribution (Zipf, 1949). For example, in written English text, the empirical distribution over words (treated as a categorical variable) can be modelled as Zipf($\alpha$, $\beta$) = Zipf(1, 2.7).

We speculate that the Preferential Attachment Theory explanation (Zhu et al., 2018; Chen, 2012) for the tendency of words to be Zipf-distributed, which states that frequently used words (features) tend to be used more often, may be analogously applicable here. At initialisation, there is some variance in feature prevalence. The tokens which are initially most highly activated early in training receive the most gradient signal and are most refined, leading to a virtuous cycle where they are more effective and useful for a larger part of the feature density spectrum. Figure 6 illustrates this dynamic over time. We would be excited about further work detailing an explicit mechanism for why features in Neural Networks tend towards being Zipf distributed.

### 7.2 ADAPTIVE COMPUTATION FOR VARYING TOKEN DIFFICULTY

One benefit of the Mutual Choice and Feature Choice approaches is that they allow for *Adaptive Computation* - difficult to reconstruct tokens (which may represent complex or rare concepts) can be reconstructed using more features, whereas other tokens which are more straightforward can use fewer features. Where Token Choice (TopK) SAEs suggest that all tokens are equal; MC and FC SAEs suggest that, in fact, some tokens are more equal than others Singer (2017).

As an extreme example, we might expect that the activations resulting from the $< \mathtt{BOS} >$ token are relatively easy to reconstruct, considering they are both very common and have exactly the same value every time. We might expect that an effective SAE could learn to productively reallocate the sparsity budget that would have been spent on the $< \mathtt{BOS} >$ token to more difficult tokens, thus increasing the SAE's effective capacity. We provide an example of the features per token distribution in Appendix D.

## 7.3 DEAD FEATURES

Our methods, especially Feature Choice SAEs, have many fewer dead features than other approaches (typically zero) without complex and compute-intensive resampling procedures (Bricken et al., 2023). This is especially important for large SAEs where the problem of dead features is typically more significant. We hypothesize that this might be an additional reason for the improved performance: all features receive some gradient signal at every step.

We note that this simplified approach eliminates the need for complex resampling procedures. This approach suggests a simplified procedure for SAE training and model selection than previous SAEs with fixed sparsity budget.

## 7.4 LIMITATIONS

**Appeals to the Monotonic Importance Heuristic:** In Section 3.1 we defined the Monotonic Importance Heuristic (MIH) - the assumption that the importance of a feature is monotonically increasing in feature activation magnitude. We use this assumption when we choose the feature activations with the largest magnitudes with our TopK-style activation functions. TopK SAEs also implicitly assume the MIH (Monotonic Importance Heuristic). We can think of Jump-ReLU SAEs and Gated SAEs as relaxing this assumption slightly to a weak MIH. For Jump-ReLU and Gated SAEs, the importance of activations is still related to their magnitude and they still filter out any low magnitude feature activations; however, the filtering threshold varies for each feature. So Jump-ReLU SAEs do not have to make magnitude comparisons across features which may have different natural scales. It may be, however, that there are low magnitude activations which, within a certain context, are nonetheless critically important in capturing information which is useful for reconstruction and/or downstream model performance. These important but low magnitude activations are difficult to capture with our current SAE approaches [10].

Though the weight-sharing form of Gated SAEs (Rajamanoharan et al., 2024a) implicitly encodes the weak MIH prior, the non-sharing form does not. Weight-sharing Gated SAEs, however, tend to perform better. The improved performance of approaches which encode the MIH prior could be considered as evidence for the truth of the claim. Alternatively, we might note that the Monotonic Importance Heuristic acts as an inductive bias for our models. Good inductive biases often allow models to perform better at first; however, with increased scale we may not need such inductive biases and may prefer allowing the model to learn more of the solution independently (Xiao, 2024), (Sutton, 2019).

**Generalisation Across Modalities:** We tested our SAEs within the domain of language. We currently don't know to what extent our results generalise across modalities, especially to inherently continuous modalities like audio and images. We would be excited about future work applying similar techniques to Interpretability problems in a wider range of modalities.

**Evaluation:** The Mechanistic Interpretability field doesn't currently have widely agreed upon metrics for evaluating Sparse Autoencoders. Disentanglement benchmarks like Huang et al. (2024) have been proposed as well as evaluation on tasks where the ground truth is known (Karvonen et al., 2024). Developing a more comprehensive suite of benchmarks for SAEs would help us to have higher confidence in our comparisons between SAE variants.

## 8 CONCLUSION

We introduce the Feature Choice and Mutual Choice SAEs as a simple drop-in change to the sparsifying activation function in Sparse Autoencoders. We also provide a new auxiliary loss, `aux_zipf_loss`, which prevents dying features and hence allows the SAE to more fully utilize all of its features without wasting capacity.

---

[10] Standard ReLU SAEs do allow low magnitude feature activations but at the expense of failing to filter out noisy low magnitude activations, which can be seen as false positives (Rajamanoharan et al., 2024b).

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

## A  FEATURE DENSITY MOVES TOWARDS A ZIPF DISTRIBUTION THROUGHOUT TRAINING

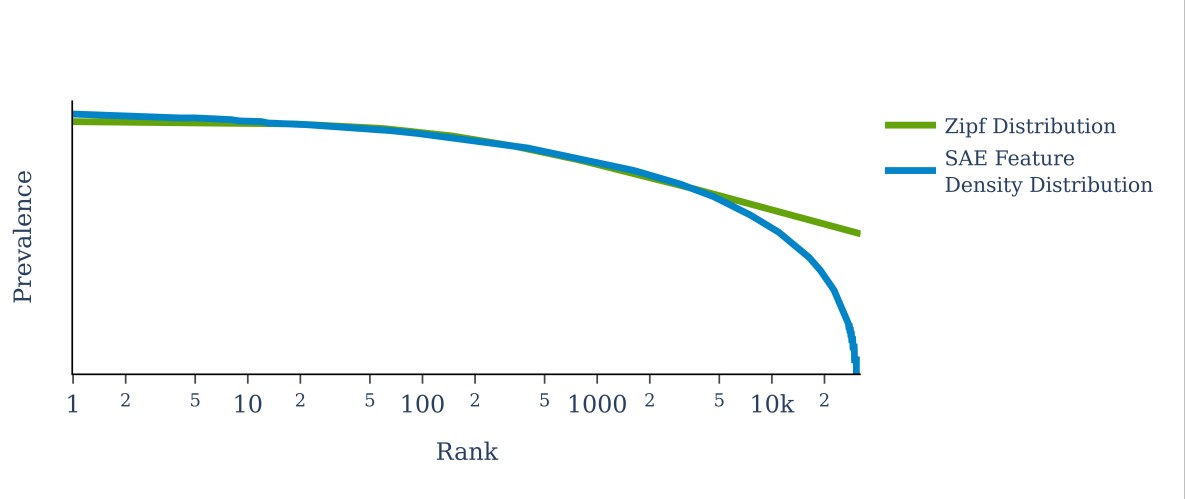

Figure 6: For an untrained SAE at initialiation, the Feature Density Distribution is less well fitted to the Zipf distribution than after training. Whilst in Figure 3, we see the Feature Distribution diverge from the Zipf distribution at the 25,000th ranked feature, here we see divergence from the 5,000th ranked feature.

Compared to after training (see Figure 3), the Feature Density Distribution diverges from the Zipf distribution at a much earlier token rank for the features at initialisation. We empirically see that over the course of training, the feature distribution approaches the Zipf distribution more closely. This convergence over training further suggests that the Zipf distribution might be a natural distribution for features. We see this pattern over multiple datasets as shown in Table 3

| Dataset | $R^2$ |
|---|---|
| Wikipedia | 0.982 |
| FineWeb | 0.983 |
| Arxiv Abstracts | 0.986 |
| Biology Arxiv Abstracts | 0.984 |
| ML ArXiv Abstracts | 0.986 |

Table 3: Across a range of datasets, the $R^2$ correlation between the Feature Density Distribution and Zipf Distribution is consistently >0.982 suggesting that the findings that features are Zipf distributed is a general phenomena.

## B  INFERENCE WITH FEATURE CHOICE AND MUTUAL CHOICE SAES

To run inference on our SAE variants, we may perform batch inference with the method exactly as in the training setup. However to do single token (or single sequence) inference (i.e. in the low batch size regime) it may be beneficial to instead impute a threshold value and swap out the activation function to use this value instead with a JumpReLU style approach (Erichson et al., 2020) (Rajamanoharan et al., 2024b).

## C  NEURAL FEATURE MATRIX LOSS

One problem with SAEs is undesirable feature splitting. Feature splitting occurs when an SAE finds a sparse combination of existing directions that allows for a smaller $L_0$ (Ayonrinde et al., 2024). For example, Bricken et al. (2023) note that a model may learn dozens of features which all represent the letter "P" in different contexts in order to maintain low sparsity.

In order to reduce feature splitting we propose two additional auxiliary losses: `nfm_loss` and `nfm_inf_loss`.

The Neural Feature Matrix (NFM) is defined as $\texttt{nfm}(\mathbf{W}) = \hat{\mathbf{W}}\hat{\mathbf{W}}^T$ for a weight matrix $\mathbf{W}$. The NFM is a symmetric square matrix which describes the correlation of different rows in the matrix $\bar{\mathbf{W}}$.

We define $\texttt{nfm\_loss} = |\texttt{nfm}(\mathbf{W}_{Dec})|_F$ and $\texttt{nfm\_inf\_loss} = \frac{1}{F}\sum_i max(\texttt{nfm}(\mathbf{W}_{Dec})_i)$.

For our largest runs we apply both of these auxiliary losses with small weight. Empirically we find this seems to reduce undesirable feature splitting and avoid a failure mode we call "Dictionary Collapse" when many features of the decoder dictionary start to align with each other. The Dictionary Collapse phenomena appears to be closely analogous to the Representation Collapse problem in Sparse Mixture of Experts (SMoE) models as detailed in Chi et al. (2022) and Do et al. (2023).

## D    ADAPTIVE COMPUTATION ALLOWS FOR A VARIABLE NUMBER OF FEATURES PER TOKEN

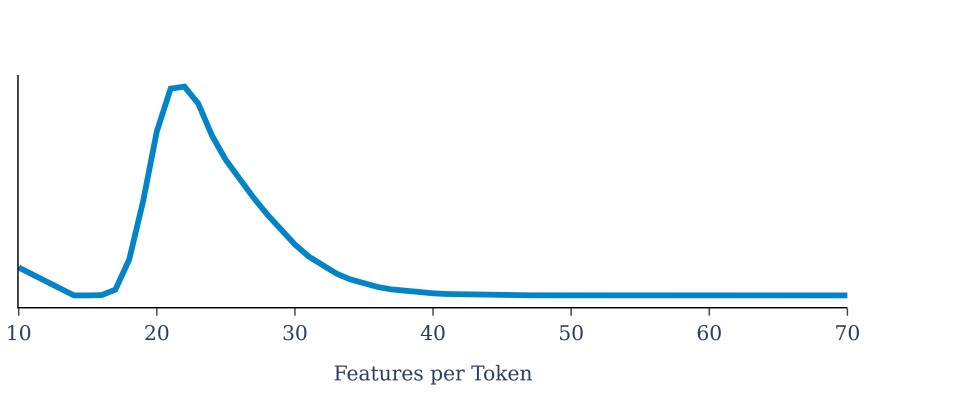

Features per Token

Figure 7: For Mutual Choice SAEs, the distribution of active features per token is bimodal with a long tail at the upper side. The SAE is able to allocate more features and more computation to more difficult to reconstruct features and perform Adaptive Computation.

We note that there is a bimodal distribution of features per token (see Figure 7). The SAE allocates close to the main mode of features to each token, attempting to allocate more features to harder tokens and less features to easier tokens. The $< \texttt{BOS} >$ token is responsible for the lower peak - this token is much easier to reconstruct and so it is prudent for the network to allocate fewer features that token. In the ideal SAE, the $< \texttt{BOS} >$ token could be reconstructed with a single feature, rather than the 10 features required here. Future work might look into optimisations that would allow the $< \texttt{BOS} >$ token to be single-feature reconstructed as a test case of allowing even greater variance in features per token.

## E    PROGRESSIVE CODES

Gao et al. (2024) describe learning a progressive code using their Multi TopK loss. In their setting, the Multi TopK loss (a weighted sum of TopK losses for different values of $k$) is required because the SAE generally "overfits" to the value of $k$ which harms a progressive code. In our case, the SAE is robust to having variable $k$ values even for the same token depending on the context of the batch. Empirically, we obtain progressive codes for a greater range of values of $k$ than in the TopK case.

## F    MONOTONIC IMPORTANCE HEURISTIC

In Section 3 we appeal to the Monotonic Importance Heuristic (MIH), as in TopK SAEs, in order to simplify our allocation problem. Empirically we find that this works well though we discuss the case for not using the MIH in Section 7.4.

We can formally write the Monotonic Importance Heuristic as the hypothesis that given some token-feature affinities $\mathbf{z'}$ and corresponding feature activations $\mathbf{z_1}$ and $\mathbf{z_2}$, if $\|\mathbf{z_1}\|_1 > \|\mathbf{z_2}\|_1$, $\mathbf{z_1}$ is likely to result in lower reconstruction error under the decoder map.

One theoretical (though informal) motivation for the MIH is as follows. Since the decoder dictionary is fixed to unit norm, the norm of any feature's contribution to the output is exactly the magnitude of the feature activation to which it corresponds. Hence if there's limited cancellation between features (which is likely in an N dimensional space where the per-token sparsity is much less than N) then we might expect the component of $\hat{\mathbf{x}}$ in any feature direction to be very close to the feature activation for that feature. In particular, consider a feature with a small magnitude of $\varepsilon$. This feature can only possibly influence the reconstruction loss by at most $\varepsilon|\mathbf{e}|$ where $\mathbf{e} = \mathbf{x} - \hat{\mathbf{x}}$. Features corresponding to larger activations can plausibly influence the reconstruction loss by a greater amount.

## G  DETERMINING THE FEATURE DISTRIBUTION

Given the Zipf exponent and bias hyperparameters, $\alpha$ and $\beta$, we use Algorithm 1 to determine the estimated feature density for each ranked feature. We use the estimated feature densities to define the threshold for dying features for the `aux_zipf_loss`.

---

**Algorithm 1** Calculate Zipf Feature Distribution

---

**Require:** $k$, $F$, $B$, $\beta$, $\alpha$, $m_{\max}$
**Ensure:** $m$: array of size $F$
1: num_interactions $\leftarrow B \times k$
2: zipf_sum $\leftarrow \sum_{i=1}^{F} \frac{1}{(i+\beta)^{\alpha}}$
3: $N_{\text{approx}} \leftarrow \frac{\text{num\_interactions}}{\text{zipf\_sum}}$
4: **for** $i = 1$ **to** $F$ **do**
5: $\quad m_i \leftarrow \min\left(\left\lfloor \frac{N_{\text{approx}}}{(i+\beta)^{\alpha}} \right\rfloor, m_{\max}\right)$
6: **end for**
7: **return** $m$

---

## H  OPTIMALITY OF TOPK ACTIVATION UNDER THE MONOTONIC IMPORTANCE HEURISTIC

**Proposition 1.** *Let $\mathbf{z} = \{z_1, z_2, ..., z_F\} \in \mathbb{R}^F$ and consider $f_z(\boldsymbol{a}) = \boldsymbol{a}^T \cdot \mathbf{z}$, where $\boldsymbol{a} = \{a_1, a_2, ..., a_F\} \in \{0,1\}^F$ is a k-sparse boolean vector and hence $\boldsymbol{a}$ satisfies $\sum_{i=1}^{F} a_i \leq k$.*

*Then for any optimal solution $\boldsymbol{a}$\*, if $a_i^* = 1$ then there exist at most $k - 1$ indices $j$ such that $z_j > z_i > 0$.*

*Proof.* We proceed by contradiction. Suppose there exists some optimal solution $\mathbf{a}$\* and an index $i$ with $a_i^* = 1$ such that there are $m > k - 1$ indices $z_j$ with $z_j > z_i > 0$. Since $\mathbf{a}$\* is k-sparse, there must exist one such index $J$ such that $z_J > z_i > 0$ and $a_J^* = 0$.

Consider $\mathbf{a}$' constructed as follows:

$$a_j' = \begin{cases} a_j^* & \text{if } j \neq i \text{ and } j \neq J \\ 0 & \text{if } j = i \\ 1 & \text{if } j = J \end{cases} \tag{5}$$

First note that $\mathbf{a}$' remains a k-sparse boolean vector as we have only swapped two elements. We see this as $\mathbf{a}$' $\in \{0,1\}^F$ and both $\mathbf{a}$' and $\mathbf{a}$\* have the same number of non-zero elements by construction.

We now note that $f_z(\mathbf{a}') - f_z(\mathbf{a}^*) = z_J - z_i > 0$, where the inequality follows from our choice of $J$. Hence we have some feasible $\mathbf{a}$' with $f_z(\mathbf{a}') > f_z(\mathbf{a}^*)$.

This contradicts our assumption that $f_z(\mathbf{a}^*)$ was optimal. $\qquad\square$

In other words, suppose we assume the Monotonic Importance Heuristic (MIH) (Section 3.1, Appendix F) on some set of real numbers, $\mathbf{z}$. If more than $k$ elements of $\mathbf{z}$ are positive, then it follows that the optimal activation function to maximise the sum of the activated affinities (and hence by the MIH, to maximise the reconstruction accuracy) under the k-sparsity constraint, is the TopK activation function. For the case where fewer than $k$ elements are positive, optimality is achieved by activating only the positive elements, as including any negative values would decrease the sum. Therefore, the composite activation function TopK $\circ$ ReLU provides the optimal solution in full generality.

