# OpenReview forum: "Adaptive Sparse Allocation with Mutual Choice \& Feature Choice Sparse Autoencoders"
_ICLR.cc/2025/Conference — Submitted to ICLR 2025_

### Official Review · Reviewer_dnCY · 2024-11-03

**Soundness:** 3
**Presentation:** 2
**Contribution:** 3
**Rating:** 6
**Confidence:** 3

**Summary:**

This paper addresses challenges in sparse autoencoders (SAEs) by introducing Feature Choice SAEs and Mutual Choice SAEs, designed to improve adaptive computation by tackling dead features and feature sparsity. These SAEs are framed as resource allocation problems via the Monotonic Importance Heuristic for adaptive feature selection, where features are adaptively assigned to tokens. Empirical validation of the proposed SAEs demonstrated that they achieve fewer dead features and improved reconstruction loss at similar sparsity levels, attributed to their intrinsic adaptive computation mechanism. Key contributions include new SAE architectures, a heuristic theoretical foundation for adaptive feature allocation, and valuable insights into efficient sparse neural models.

**Strengths:**

This work has potential for applications needing efficient feature allocation and reduced model complexity, like sparse representation learning. The adaptive computation strategy to reduce dead features could improve SAE design, making it more efficient for high-dimensional settings. Though incremental, the contributions offer valuable insights into current SAE limitations. The paper provides a basic experimental setup with empirical validation on real-world datasets. The originality of this work lies in framing SAEs as resource allocation problems and introducing the Feature Choice SAE and Mutual Choice SAE variants, offering a new approach to adaptive computation.

**Weaknesses:**

- **Lack of Reproducible Code**: The paper does not include reproducible code for the numerical experimental results, making it difficult to verify the empirical findings. Consequently, my comments are based solely on the results as presented in the paper.

- **Monotonic Importance Heuristic:** The theoretical framing of models such as TopK SAEs, Feature Choice SAEs, and Mutal Choice SAEs as resource allocation problems is coherent. However, it is important to verify the mathemtical rigor in their proofs, even though the authors offer an informal theoretical motivation in Appendix G.

- **Lack of Extensive and Comprehensive Experimental Setup**: Although the comparisons with standard SAEs aim to demonstrate improvements in dead features and reconstruction loss, the evaluation lacks thoroughness and would benefit from a more comprehensive approach (see additional details in Questions).

- **ICLR 2025 Citation Style**: The paper should adhere to the ICLR 2025 citation style: when the authors or publication are part of the sentence, the citation should appear without parentheses using \citet{} (e.g., “See Hinton et al. (2006) for more information.”). Otherwise, the citation should be placed in parentheses using \citep{} (e.g., “Deep learning shows promise to advance AI (Bengio \& LeCun, 2007).”) The authors should revise all citations to ensure they adhere to this standard format.

- **Inconsistent Mathematical Notations**: The notation for multivariate variables in equations (1), (2), (3), and beyond is inconsistent, alternating between bold and regular font. Following a consistent notation style, as recommended by the ICLR guidelines, would enhance clarity.

- **Missing Information on Citations**: There are several instances of missing conference or journal names in the references. The authors should address these omissions to ensure completeness. For example, on lines 648-650, the correct citation format should be: "William Fedus, Barret Zoph, and Noam Shazeer. Switch Transformers: Scaling to Trillion Parameter Models with Simple and Efficient Sparsity, 2022. *Journal of Machine Learning Research*, 23.120 (2022): 1-39."

- **Typos:**   **Line 206**: The mask $S$ should be written in bold as $\mathbf{S}$. **Line 266:** aux\_$k_l$oss should be aux\_k\_loss.  **Line 903:** "Algorithm 1 algorithm 1" should be "Algorithm 1". **Line 452:** "fig. 6" should be "Fig. 6".

**Questions:**

- **Lack Comparison with Concurrent Work:** In addition to the baseline SAE types listed in Table 2, the authors should include a comparison with Bussmann et al. (2024) [1], which introduced BatchTopK-a method that also offers adaptive computation and could serve as a useful benchmark agaist Mutual Choice SAEs.

References:

[1] Bart Bussmann, Patrick Leask, and Neel Nanda. BatchTopK: A Simple Improvement for TopK-SAEs. July 2024. https://www.alignmentforum.org/posts/Nkx6yWZNbAsfvic98/batchtopk-a-simple-improvement-for-topk-saes

- **Line 195**: Could the authors clarify the definition of the pre-sparsifying activation feature magnitudes $\mathbf{z}'$? I assume this refers to the token-feature affinities, but how is this value determined for real applications?

- **Line 258-260**: The authors should provide a more detailed definition of "dying features" to improve clarity.

- **Table 1**: Could the authors clarify the term "w/ resampling"? Is it a fair comparison to evaluate results with and without resampling? Additionally, in practical applications, which approach yields a lower percentage of Dead Features?

- **Line 855-857:** The authors mentioned the failure mode "dictionary Collapse", which I personally think it is analogous the the collapsing problem in sparse mixture of experts models. The authors should demonstrate this via visualization or quantitative empirical results.

- **Line 855-857**: The authors mention the failure mode "dictionary collapse," which I believe is analogous to the collapsing problem seen in sparse mixture of experts models [1,2]. To strengthen this point, the authors should consider demonstrating it through visualizations or quantitative empirical results as in [1,2].

References:

[1] Chi, Z., Dong, L., Huang, S., Dai, D., Ma, S., Patra, B., Singhal, S., Bajaj, P., Song, X., Mao, X.L. and Huang, H., 2022. On the representation collapse of sparse mixture of experts. Advances in Neural Information Processing Systems, 35, pp.34600-34613.

[2] Do, T.G., Pham, Q., Nguyen, T., Doan, T.N., Nguyen, B.T., Liu, C., Ramasamy, S., Li, X. and Steven, H.O.I., HyperRouter: Towards Efficient Training and Inference of Sparse Mixture of Experts. In The 2023 Conference on Empirical Methods in Natural Language Processing.

---

> ### Author Response · Authors · 2024-11-30
>
> We thank the reviewer for their feedback and comments.
>
> We agree with the reviewer that our work details some core limitations of current SAEs, provides a framework for understanding and improving SAEs and reduces dead features allowing SAEs to scale to much larger dimensional problems.
>
> We thank the reviewer for their comments on previous weaknesses and questions which we address in turn:
>
> Reproducible Code: We have now made our code available to the community to ensure full reproducibility.
>
> Monotonic Importance Heuristic (MIH): We have now added a further definition of the MIH in Appendix F (lines 916-917) and we added a proof that the optimal activation function given the MIH is the TopK activation function that we propose in Appendix H. This adds mathematical rigour and can be combined with the intuition of the motivating argument in Appendix G (which is now Appendix F).
>
> Comparison with Concurrent Work: As the reviewer suggests, we have added a comparison to Bussman et al.'s current work in Table 2.
>
> Citation Style and Information in Citations: We have revised all citations to ensure they adhere to the advised format and have the most up to date conference and journal names.
>
> Mathematical Notation: On the guidance of the reviewer, we have updated the mathematical notation to be consistent across the paper using bold face for tensor quantities.
>
> Typos: We thank the reviewer for noticing and alerting to us to these three typos which have now been amended.
>
> Questions:
>
> 1. We have added a definition of the token-affinities z' in line 109 and have been more consistent with the notation to reflect this. Briefly, given some neural activations x, we first apply the encoder which returns z', the dense token-feature affinities. We then apply the activation function to get z' in order to get z, the feature activations corresponding to x. The token-feature affinities that we refer to can be thought of as the feature activations before the sparsifying activation function is applied.
>
> 2. We have added a more explicit definition of "dying features" in Section 3.2, Definition 3.1.
>
> 3. Resampling yields a lower percentage of dead features so we include the comparisons with resampling such that we are comparing against a stronger baseline. We add some further details in lines 340-342 and Footnote 7 which we reproduce here: "Bricken et al. (2023) describe a resampling procedure for dead features in which they reinitialise these dead features midway through training in order to reduce the number of dead features at the end of training. Models with the aux_k_loss do not require resampling but for models without the aux_zipf_loss we include resampling for stronger baselines."
>
> 4. We thank the reviewer for pointing out the close analogy between our notion of dictionary collapse and the collapsing problem in sparse mixture of experts models. We have added some additional detail to this section and more clearly made the comparison in Appendix C and cited the relevant work.
>
> We again thank the reviewer for their comments. Given these clarifications and additional experiments demonstrating broader generalisation across architectures and datasets, we kindly ask if the reviewer is willing to reevaluate their score.

---

### Official Review · Reviewer_YRih · 2024-11-04

**Soundness:** 3
**Presentation:** 2
**Contribution:** 3
**Rating:** 5
**Confidence:** 3

**Summary:**

The paper introduces two novel variants of Sparse Autoencoders (SAEs) - Feature Choice SAEs and Mutual Choice SAEs - for extracting interpretable features from neural networks. The key innovation is framing the sparsifying activation function as a resource allocation problem, where a fixed sparsity budget must be distributed across token-feature matches. The authors also introduce aux_zipf_loss, which addresses underutilized features by generalizing the existing aux_k_loss. The proposed methods show improvements in reconstruction accuracy and feature utilization compared to existing approaches like TopK SAEs, particularly in handling dead features.

**Strengths:**

1. The resource allocation perspective on sparsifying activation functions provides an intuitive framework for understanding and improving SAEs.
2. The Feature Choice and Mutual Choice variants offer clear improvements over existing methods. For example, the Feature Choice SAEs reliably achieve zero dead features even at large scale and the scaling results are good that is able to handle up to 34M dimension models.
3. The discussion on the Zipf distribution in feature usage is also thoughtful.

**Weaknesses:**

1. One of my main concerns is about the presentation. Some technical details are scattered throughout different sections and the Figures and Tables are not clear or elegant which are beyond the width of the paper. The figures could also be better explained. The paper would benefit from clearer organization and improved writing.

2. The experiment results are somewhat not sufficient. Experiments primarily focus on GPT-2 small. More diverse model architectures would strengthen claims.

3. The contribution of this work is somehow incremental with limited significance and impact. The core problem about dead features in SAEs handled by this paper has limited impact in the interpretability community.

**Questions:**

1. The paper claims improved reconstruction accuracy, but how does this translate to downstream task performance beyond the provided loss metrics?
2. Could the authors elaborate on the computational overhead of Feature Choice and Mutual Choice SAEs compared to TopK SAEs?
3. How well do these methods generalize to other types of neural networks beyond language models?
4. What are the practical implications of these improvements for real-world applications?

---

> ### Author Response · Authors · 2024-11-30
>
> We thank the reviewer for their comments and queries.
>
> In response to the reviewer's main concern we have restructured the paper and ensured that the technical details relevant for the method are present in the Method section. We have also updated the captions for all figures, improved writing where it was unclear (for example unifying notation and adding an explicit definition of the token-feature affinities [line 109], the Monotonic Importance Heuristic [lines 916-917 and 187-189] and Dying Features [Definition 3.1]) and ensured that all figures and tables are an appropriate width.
> We also remade all figures to increase their clarity.
>
> We also added experiments with several different datasets in Table 3 to corroborate our results and clarified that the 16M and 34M SAE width experiments in Table 1 were conducted with Pythia-2.8B-deduped which is both a different and much larger underlying model. We also conducted experiments with 4 different SAE architectures with 3 ablation types across varying sparsity hyperparameters and 5 different model sizes varying over 4 orders of magnitude.
>
> Response to questions:
>
> Questions 1 & 3: We have added some additional discussion in Section 7.4 detailing that the generalisation of these methods to domains outside of language and applications is outside the scope of this work. We would be interested in follow-up work that focused on examining the generalisation to other domains and we might point the reviewers to Fry et al. (2024)[^1] and Gorton (2024)[^2] as examples of early work in this direction.
>
> Question 2: We have added lines 455-488 to confirm that as in line 88, we are comparing to TopK SAEs with our approaches in a FLOP-matched way. There is no additional computational overhead and empirically we do not observe wall-clock slow downs despite not having optimised for wall-clock time specifically.
>
> Question 4: With regards to the practical implications of this work for real-world applications and the impact on the interpretability community we respectfully disagree with the reviewer that this work has limited significance and impact. Having a scalable and accurate way to decompose the activations of foundation models into linear combinations of meaningful, understandable and monosemantic features is useful for applications in debugging model performance for improving foundation models, having more fine-grained control of models without side-effects for reducing harmful and biased outputs in widely deployed foundation models, and in uncovering latent scientific principles that may underly model performance for advancing the natural sciences. Despite SAEs being promising in achieving these aims, they are currently not sufficiently accurate to produce impacts I described above. We believe that increasing the accuracy of SAEs is a likely path to enabling SAEs to achieve these technical, social and scientific aims.
> In addition to our work presenting a contribution to improving SAEs we also give a framework for understanding and improving SAEs which the reviewer notes is intuitive. We believe other researchers may be able to utilise this framework to further improve the efficacy of feature extraction and feature-based interpretability.
>
> We thank the reviewer for their suggestions which contributed to improving this work. Conditional on the reviewer agreeing that we have addressed their main concerns about presentation and the lesser concerns about the diversity of experiments and impact on the interpretability community, we would kindly ask the reviewer to reconsider their assessment of our work's contribution.
>
> [^1]: https://arxiv.org/abs/2410.03334
> [^2]: https://arxiv.org/abs/2406.03662

---

> > ### Comment · Reviewer_YRih · 2024-12-02
> > **Thank you for your rebuttal**
> >
> > Thank you for your rebuttal. While I appreciate your efforts to address my concerns, several key issues remain:
> >
> > 1. The presentation improvements are difficult to verify without highlighted changes. Simply shrinking PDF margins is insufficient and not allowed. I would strongly prefer to see a version with color-coded revisions to properly evaluate the changes.
> >
> > 1. While you've added more experiments, including Table 3 with different datasets, the validation still primarily focuses on language models. The practical benefits of solving the dead features problem need stronger demonstration.
> >
> > 1. Your responses to Questions 1 and 3 about generalization and downstream performance mostly defer to future work, which doesn't fully address my concerns about immediate practical impact.
> >
> > Given these remaining issues, particularly regarding presentation and practical impact demonstration, I maintain my original assessment that the paper falls marginally below the acceptance threshold. I encourage you to provide a version with highlighted changes and include more concrete examples of practical benefits.

---

> > > ### Author Response · Authors · 2024-12-04
> > >
> > > We thank the reviewer for their comments.
> > >
> > > To see the presentation improvements, as per the guidance of ICLR-2025, the *pdfdiff* tool of OpenReview can be applied to compare new changes of the paper against the original submission.
> > > For changes to the figures each of the figures from Figures 1-7 have been updated for additional clarity, in accordance with your review as well as Tables 1-2 (Table 3 was added). It is unfortunately not possible to provide a version with color-coded, highlighted changes as the pdf is no longer editable but we thank you for the idea for the future.
> > >
> > > We respectfully disagree with the comments that validation only on language models doesn't demonstrate our findings. It is the standard across papers in this field to test on a single modality and other papers at this conference and in other venues have not been held to the standard that they must compare across modalities (e.g. [1], [2], [3]). However in response to your valuable feedback we will produce experiments on vision tasks for the camera-ready version of this work.
> > >
> > > We again thank the reviewer for their comments.
> > >
> > > [1]: https://openreview.net/forum?id=tcsZt9ZNKD
> > > [2]: https://openreview.net/forum?id=F76bwRSLeK
> > > [3]: https://neurips.cc/virtual/2024/poster/92961

---

### Official Review · Reviewer_nZSN · 2024-11-04

**Soundness:** 2
**Presentation:** 2
**Contribution:** 3
**Rating:** 6
**Confidence:** 3

**Summary:**

The paper proposes two new variants of sparse autoencoders (SAEs), namely Feature Choice SAEs and Mutual Choice SAEs, to address the limitations of existing TopK SAEs. A key improvement is that the proposed models adaptively allocate the total sparsity budget between tokens and features, resulting in improved reconstruction performance (Figure 4). Additionally, the paper introduces a new auxiliary loss function, aux_zipf_loss, to mitigate underutilized features based on the observation that the Zipf distribution generally fits most of the feature densities, except at the lower end. The proposed aux_zipf_loss enhances feature utilization and reduces dead features (Table 1).

**Strengths:**

Clear Visualization: Figure 1 effectively visualizes the design of the sparsifying activation functions and the differences between the proposed methods and existing approaches.

Adaptive Computation: Feature Choice SAE and Mutual Choice SAEs demonstrate improved performance through adaptive computation, effectively allocating more resources to more complex tokens.

Mitigating Dead Features: The introduction of the aux_zipf_loss and the proposed definition of dying features successfully mitigate the occurrence of dead and underutilized features, leading to better overall model performance.

**Weaknesses:**

Organization of Sections: Sections 3 and 4 appear somewhat disorganized, with some concepts in Section 3 that would be better suited for the Methods section. This restructuring could improve the overall flow and clarity of the paper.

Limited Evaluation Scope: The analysis of the Zipf distribution fit (Figure 2) is an interesting discovery, but it appears that the authors only conducted this analysis on GPT-2. Broader validation on different models or datasets would strengthen the claims.

**Questions:**

1. In Line 344, the authors mentioned that they evaluated the interpretability of the SAEs using Juang et al. (2024)'s automated interpretability (AutoInterp) process. Could the authors provide results of this evaluation?

2. Could the authors provide more details about the experimental setup, including the datasets and the exact procedures used, to ensure the reproducibility of the results? Ideally, releasing the codebase would greatly benefit the community.

3. Have the authors consistently observed the discrepancy between the feature density and the Zipf curve in different settings, or is this observation limited to the GPT-2 experiments presented here?

---

> ### Author Response · Authors · 2024-11-30
>
> We thank the reviewer for their comments and questions.
>
> We agree with the reviewer that our approach leads to overall model performance due to the use of adaptive computation and the reduction of dead features. We thank the reviewer for their praise of Figure 1 in comparing the methods and we have further improved this figure for additional clarity.
>
> Organisation of Sections: We thank the reviewer for their comments on the structuring of the sections. In response we have now moved the definition of our activation functions, the auxiliary loss and the zipf constraint to the Methods section to improve the flow of the paper as the reviewer suggests.
>
> Evaluation Scope: We have now included a comparison of the Zipf distribution fit for 5 other datasets in Table 3 to corroborate our findings. We agree with the reviewer that this increased diversity strengthens the claim. We also clarify that the 16M and 34M experiments in Table 1 are conducted with Pythia-2.8B-deduped which also shows validation of the Zipf Feature Choice strategy on models of different architectures and sizes.
>
> Experimental Setup: We have added additional details about the experimental setup including the datasets and procedures in lines 335-341. The main dataset that we use is FineWeb and we provide experiments with additional datasets in Table 3. We also release the codebase as the reviewer suggests which ensures greater reproducibility of the results.
>
> Discrepancy between Feature Density and Zipf curve: We have included additional experiments in Table 3 as above which show that this observation is not limited purely to a single model. We also added additional detail to the discussion in Appendix A describing that we see the discrepancy between the two curves being larger at the start of training and decreasing over the course of training, as the feature density converges towards a Zipf distribution with increased training.
>
> Given these clarifications and additional experiments demonstrating broader generalisation across architectures and datasets, we kindly ask if the reviewer is willing to reevaluate their score.

---

> > ### Comment · Reviewer_nZSN · 2024-12-03
> >
> > I appreciate the authors’ response and the additional experiments. I have raised my points by 1 level since my concerns are resolved.

---

> > > ### Author Response · Authors · 2024-12-04
> > >
> > > We thank the reviewer for their comments. Given that the updates to the paper increase the presentation and scope, would the reviewer consider reevaluating their presentation and soundness scores?

---

### Official Review · Reviewer_cuu5 · 2024-11-04

**Soundness:** 3
**Presentation:** 2
**Contribution:** 2
**Rating:** 5
**Confidence:** 2

**Summary:**

The paper proposed Feature Choice and Mutual Choice Sparse Autoencoder (SAE). The feature choice version enforces sparsity by restricting each feature to match with at most $m$ tokens; the mutual choice version allows any choice of token-feature matching. zipf distribution is used to set the constraints for feature choice, and an additional auxiliary loss is used to prevent dead features. Numerical experiments show the proposed methods outperform existing SAE methods in terms of reconstruction at different sparsity levels and percentage of dead features

**Strengths:**

The proposed methods effectively reduce dead features and allow for flexible feature allocations, which allows the model to utilize features better and potentially more interpretable.

**Weaknesses:**

1. The experiments are rather limited, only one setting with GPT2 activations is performed.
2. To demonstrate the significance of the results, I think it would be better to include the average and standard deviation of the evaluated metrics across different runs
3. SAE is described as a promising approach for extracting sparse, meaningful, and interpretable features from neural networks. But under the current experiment setup, it is hard to tell how the extracted features are meaningful and interpretable.
4. The problem is framed as a resource allocator in section 3, but I think it is not very clear how the edge weights are defined. If I understand correctly, it is somehow defined as related to the magnitude of feature activation, e.g. in line 483.
5. There are some typos, e.g. in lines 210 and 221 $S$ is defined the same way, but shouldn't it be different?  in line 266, l is written in as a subscript in aux_k_loss

**Questions:**

See weakness

---

> ### Author Response · Authors · 2024-11-30
>
> We thank the reviewer for their comments and suggestions.
>
> In response we have improved the presentation of several parts of the paper including figures, graphs and captions.
>
> We have also addressed all the previous weaknesses that the reviewer points out:
> 1. We have added experiments with multiple different datasets and to corroborate the results as seen in Table 3 and we have clarified that experiments on 16M and 34M width SAEs (Table 1) were conducted on Pythia-2.8B-deduped which shows that our methodology generalises to larger models with different architectures.
>
> 2. As above we have now included a comparison with multiple runs across multiple datasets in Table 3.
>
> 3. We agree that we describe SAEs as extracting sparse, meaningful and interpretable features. As to the question of whether the features extracted are meaningful, we argue that since a linear combination of a small number of features explains a large fraction of the variance (as measured by the normalised loss recovered metrics for example), that suggests that the features are meaningful.
>
> 4. We thank the reviewer for pointing out that this wasn't clear, we have added some further exposition on line 109. Given some neural activations x, we first apply the encoder which returns z', the dense token-feature affinities. We then apply the activation function to get z' in order to get z, the feature activations corresponding to x. The token-feature affinities that we refer to can be thought of as the feature activations before the sparsifying activation function is applied. We have also unified our naming convention to refer to these explicitly as token-feature affinities.
>
> 5. We thank the reviewer for pointing out the typo on line 266 which is now fixed. We have also fixed the discrepancy between (now) lines 197 and 204 where the dimensions were previously the same.
>
> We hope these additional changes and clarifications have addressed the reviewer's concerns. Please let us know if there are additional questions. Given these clarifications and additional experiments demonstrating broader generalisation across architectures and datasets, we kindly ask if the reviewer is willing to reevaluate their score.

---

> > ### Author Response · Authors · 2024-12-04
> >
> > We believe that we have addressed the concerns that the reviewer had, are there any further comments from the reviewer in response?

---

### Meta-Review · Area_Chair_6t8Y · 2024-12-05

**Metareview:**

This work considers the token-feature matching problem derived from sparse autoencoder. The authors cast it into resource allocation framework, and considered two variants distinguished by their own constraints. Empirical results show that the methods result in more active feature participation and lower reconstruction loss.

The reviewers agree that this is an interesting work, and is well motivated.

However, the reviewers raised concerns that the empirical study is insufficient to justify the main claim, due to testing on small scale models and lack of important evaluation metric such as mean and standard deviation. In addition, reviewers also found the paper is not well organized.

**Additional Comments On Reviewer Discussion:**

Reviewers shared the concern that the empirical study is insufficient. The Area Chair agrees with them.

---

### Decision · Program_Chairs · 2025-01-22

Reject